# Influence of transurethral enucleation with bipolar of the prostate on erectile function: Prospective analysis of 51 patients at 12-month follow-up

**Yasuyuki Kobayashi●\*, Hiroki Arai, Masahito Honda**

Department of Urology, Kinki Central Hospital of Mutual Aid Association of Public School Teachers, Itami, Hyogo, Japan

\* ya_su_koba@yahoo.co.jp

## Abstract

### Background

Transurethral enucleation with bipolar (TUEB) is a safe and effective surgery for benign prostatic obstruction (BPO). However, few data exist concerning the influence of TUEB on erectile function (EF) in patients with BPO.

### Objective

To evaluate the influence of TUEB on EF in patients with BPO at 3- and 12-month follow-up.

### Material and methods

We prospectively enrolled 51 patients who underwent TUEB from June 2016 to April 2020. We evaluated maximum urinary flow rate (Qmax), postvoid residual urine (PVR), International Prostate Symptom Score (IPSS), quality of life (QoL), and International Index of Erectile Function-5 (IIEF-5) preoperatively and at 3- and 12-month follow-up. We classified the patients according to their preoperative IIEF-5 score into group 1 (IIEF-5 $\geq$10; n = 24) and group 2 (IIEF-5 <10; n = 27), and for further evaluation of EF, into subgroups a: severe (IIEF-5 5–7), b: moderate (8–11), c: mild to moderate (12–16), d: mild (17–21), and e: no erectile dysfunction (22–25). Data are displayed as median or median (interquartile range).

### Results

The study comprised 51 patients with a median age of 75 (70.5–79.5) years. Median prostate and transition zone volumes were 69.5 (46.5–78.8) mL and 30.5 (19–43) mL, respectively. Urinary function improved significantly when comparing respective preoperative, 3-month, and 12-month follow-up values: Qmax (7.6, 12.9, 15.2 mL/s), PVR (50, 0, 0 mL), IPSS (20.5, 9, 6), and QoL (5, 2, 2), respectively. There was no significant change in IIEF-5 score across the three time points: 9, 7, 8. The IIEF-5 score slightly but significantly increased between the preoperative and 12-month follow-up values in group 2 (5, 5, 6) and subgroup a (5, 5, 5).

**Data Availability Statement:** All relevant data are within the paper and its Supporting Information files.

**Funding:** The authors received no specific funding for this work.

**Competing interests:** The authors have declared that no competing interests exist.

## Conclusion

TUEB was effective and safe surgery for patients with BPO and showed no significant influence on EF at 12-month follow-up after TUEB in patients with BPO.

## Introduction

Benign prostatic hyperplasia (BPH) is a histological diagnosis regarding the proliferation of glandular epithelial tissue, connective tissue, and smooth muscle in the prostate transition zone (TZ) [1]. This condition can progress through benign prostatic enlargement to benign prostatic obstruction (BPO) [2]. BPO is a subset of bladder outlet obstruction and is diagnosed when the cause of obstruction is thought to be benign prostatic enlargement [2]. Monopolar transurethral resection of the prostate (TURP) has been considered the gold standard surgical treatment for BPO [3]. In recent years, a variety of transurethral surgical treatments have been developed to achieve comparable surgical outcomes and to reduce complications. In addition to monopolar TURP, bipolar TURP, holmium laser enucleation of the prostate (HoLEP), and photoselective vaporization of the prostate (PVP) have become the main surgical procedures for patients with BPO [4–6]. Recently, transurethral enucleation with bipolar (TUEB) surgery has been developed, which is a transurethral bipolar enucleation method using a specially designed loop for enucleation and coagulation (TUEB loop) [7]. Although the benefits of TUEB have not been established against TURP, previous studies reported high efficacy and safety and a low perioperative morbidity rate with TUEB [7–11]. When surgery is considered for patients with BPO, erectile function (EF) is one of the considerable surgical factors [12]. There were several reports that TURP (monopolar or bipolar), HoLEP, and PVP had no significant influence on EF [13–15]. Although a majority of studies showed that transurethral surgery for BPO has no significant influence on EF, few data exist concerning the influence of TUEB on EF [12]. Therefore, we conducted a single-center, prospective study to evaluate the influence of TUEB on EF in 51 patients with BPO.

## Material and methods

This prospective study investigated patients with BPO who underwent TUEB from June 2016 to April 2020. The Ethics Committee of our institution approved this study on 23 May 2016 (approval no. 290) and 29 October 2018 (approval no. 363). This study complied with the Declaration of Taipei on Ethical Considerations regarding Health Databases and Biobanks [16]. We obtained written informed consent from all patients who participated in this study. The authors ascertain the availability of all original data reported in this study.

Eligible patients were aged 20 years or older and required surgery for BPO. We consider TUEB to be the standard surgery in patients with BPO refractory to medical therapy (including urinary retention) in our institution. We excluded patients with severe urethral stricture requiring urethrotomy, previous prostate surgery, or a history of bladder cancer or prostate cancer. We prospectively enrolled 51 BPO patients who underwent TUEB from June 2016 to April 2020 and were followed up at 3 and 12 months postoperatively. We preoperatively performed a general clinical evaluation with digital rectal examination, urinalysis, maximum urinary flow rate (Qmax), postvoid residual urine (PVR), International Prostate Symptom Score (IPSS), quality of life (QoL) and International Index of Erectile Function-5 (IIEF-5) scores, serum prostate-specific antigen (PSA), transabdominal ultrasonography, cystoscopy, and

pelvic magnetic resonance imaging (MRI). Only one patient who was unable to undergo MRI due to contraindications to MRI was excluded from the prostate volume analysis. We collected preoperative and postoperative data including operation time and length of time catheterized. At the 3- and 12-month follow-ups, we collected questionnaires (IPSS, QoL, and IIEF-5) and measured Qmax and PVR.

We measured prostate volume and TZ preoperatively with pelvic MRI. We performed transverse measurements in the axial plane, which shows the maximal diameter and allows best visualization of the surgical capsule and enlarged TZ boundaries for transverse measurement. We also performed sagittal length measurement in the sagittal plane, which shows the urethra most clearly [17]. We calculated the volume by using the maximal height and width in the axial plane, and the length of the prostate and TZ in the sagittal plane, applying the formula for a geometric model of an oblong ellipsoid [17].

TUEB was performed by two surgeons (Y.K. and H.A.) in our department with the patient under general anesthesia in a lithotomy position. We used a bipolar generator (Olympus Surg-Master UES-40), TUEB loop, standard wire loop, and 26 Fr resectoscope (all from Olympus, Tokyo, Japan). The TUEB loop comprised two parts, a front-end polytetrafluoroethylene loop designed for blunt enucleation (spatula) and a standard wire loop for coagulation. The generator for TUEB was set at 280 W for cutting and 100 W for coagulation. We used normal saline (0.9%) as irrigation fluid. After confirming the bilateral ureter orifice, bladder, verumontanum, and sphincter, we marked the resection borders at the proximal part of verumontanum from the 5 to 7 o'clock position to gain the enucleation plane. Then, we marked the resection borders circumferentially. We found the smooth plane with clear vessels between the adenoma and the capsule at 5 to 7 o'clock and enucleated the adenoma from the capsule with the TUEB loop. If bleeding occurred, we immediately coagulated it with the TUEB loop. If the middle lobe existed, we enucleated the adenoma from the capsule retrogradely toward the bladder neck. We sequentially enucleated the adenoma bilaterally and anteriorly. After performing subtotal enucleation of the adenoma and leaving a bridge of tissue at the bladder neck at the 6 o'clock position, we resected the adenoma layer by layer with a standard wire loop with little to no bleeding. After we resected the bridge of tissue and any residual adenoma, we evacuated the prostate tips with a bladder syringe and ensured complete hemostasis. We performed lithotripsy at the same time in the patients with bladder stones. At the end of surgery, we inserted a 22 Fr 3-way Foley catheter with continuous irrigation. We discontinued irritation on the morning after surgery. The catheter was removed 4 to 6 days after TUEB according to the study protocol. We checked hemoglobin before and the day after surgery. We noted all surgical complications until the 12-month follow-up and classified them according to the Clavien-Dindo classification system [18].

According to preoperative IIEF-5 scores, we classified the patients into group 1 (IIEF-5 ≥10, n = 24) and group 2 (IIEF-5 <10, n = 27). For the further evaluation of EF, we also classified the patients according to their preoperative IIEF-5 score into subgroups a: severe (IIEF-5 5–7, n = 23); b: moderate (IIEF-5 8–11, n = 13); c: mild to moderate (IIEF-5 12–16, n = 9); d: mild (IIEF-5 17–21, n = 5); and e: no erectile dysfunction (IIEF-5 22–25, n = 1) [19].

Data are reported as the median or median (interquartile range). We compared the categorical variables using Fisher's exact test and calculated the differences between each group using the Mann-Whitney U test and Kruskal-Wallis test. We calculated the changes of the parameters in each group preoperatively and at the 3- and 12-month follow-ups using the Wilcoxon signed rank test. We assessed the factors that influence the IIEF-5 score using logistic regression. The data were analyzed as of 1 June 2021. All statistical analyses were performed using the open-source software EZR version 1.27 (Saitama Medical Center, Jichi Medical University, Saitama, Japan), which is a graphical user interface of R (The R Foundation for Statistical

Computing, Vienna, Austria). P-values are two-sided, and a value of <0.05 was considered statistically significant.

## Results

We registered 51 patients with BPO in the study and performed TUEB on all of them. Their characteristics are listed in Table 1. The median age was 75 (70.5–79.5) years, and that of group 1 was significantly younger than that of group 2 (72 vs 76 years). Median prostate volume and TZ volume were 69.5 (46.5–78.8) mL and 30.5 (19–43) mL, respectively, with no significant differences between the two groups. Nine (17.6%) patients on an alpha-blocker had urinary retention and could not void without catheterization when we performed TUEB. Twenty-four (47.1%) patients had hypertension, 12 (23.5%) had diabetes mellitus, and 10 (19.6%) had cardiovascular disease. The patients with cardiovascular disease were all in group 2.

The perioperative data of this study are listed in Table 2. TUEB surgery was completed successfully in all patients. There were no statistically significant differences in the perioperative data between groups 1 and 2. The complications (according to the Clavien-Dindo classification) associated with TUEB are shown in Table 3.

**Table 1. Patient characteristics.**

| | All men (n = 51) | | Patients with IIEF-5 ≥10 (Group 1; n = 24) | | Patients with IIEF-5 <10 (Group 2; n = 27) | | p Value |
|---|---|---|---|---|---|---|---|
| Age (y) | 75 | (70.5–79.5) | 72 | (68.5–75) | 76 | (72–80.5) | 0.0357 |
| PSA (ng/mL) | 5.8 | (2.6–10.1) | 7.4 | (3.9–10.6) | 3.7 | (2.0–7.6) | 0.0713 |
| Prostate volume (mL) | | | | | | | |
| Total | 69.5 | (46.5–78.8) | 70.5 | (63.8–78) | 53 | (38.2–95.5) | 0.356 |
| Transition zone | 30.5 | (19–43) | 31 | (25.3–40) | 27.5 | (14.8–44.5) | 0.443 |
| Urinary retention before TUEB, n (%) | 9 | (17.6) | 4 | (16.7) | 5 | (18.5) | 1 |
| IPSS | 20.5 | (16.8–28) | 18.5 | (14.2–22) | 24 | (18.3–30) | 0.0266 |
| QoL | 5 | (4–6) | 5 | (4–6) | 5 | (4–6) | 0.991 |
| Qmax (mL/s) | 7.6 | (5.8–10.9) | 6.9 | (5.4–10.6) | 8.4 | (6.5–11.2) | 0.174 |
| PVR (mL) | 50 | (3–107.8) | 53 | (22.5–94.8) | 42 | (3–107.8) | 0.98 |
| IIEF-5 | 9 | (5–12.5) | 13.5 | (10.8–15.5) | 5 | (5–6) | <0.001 |
| Prostate morphology, n (%) | | | | | | | |
| Bilateral lobes enlarged | 42 | (82.4) | 18 | (75) | 24 | (88.9) | 0.276 |
| Bilateral and middle lobes enlarged | 9 | (17.6) | 6 | (25) | 3 | (11.1) | 0.276 |
| Bladder stone, n (%) | 3 | (5.9) | 2 | (8.3) | 1 | (3.7) | 0.595 |
| Alpha-blocker therapy, n (%) | 50 | (98) | 24 | (100) | 26 | (96.2) | 1 |
| Alpha-reductase inhibitor therapy, n (%) | 5 | (9.8) | 2 | (8.3) | 3 | (11.1) | 1 |
| ASA score | 2 | (2–2) | 2 | (2–2) | 2 | (2–2) | 0.215 |
| Past history n (%) | | | | | | | |
| Hypertension | 24 | (47.1) | 8 | (33.3) | 16 | (59.3) | 0.093 |
| Diabetes mellitus | 12 | (23.5) | 4 | (16.7) | 8 | (29.6) | 0.335 |
| Cardiovascular disease | 10 | (19.6) | 0 | (0) | 10 | (37) | <0.001 |

IIEF-5: International Index of Erectile Function-5, IQR: interquartile range, PSA: prostate-specific antigen, TUEB: transurethral enucleation with bipolar, IPSS: International Prostate Symptom Score, QoL: quality of life, Qmax: maximum urinary flow rate, PVR: postvoid residual urine.

Data are shown as the median (interquartile range).

**Table 2. Perioperative data.**

| | All men (n = 51) | | Patients with IIEF-5 ≥10 (n = 24) | | Patients with IIEF-5 <10 (n = 27) | | p Value |
|---|---|---|---|---|---|---|---|
| Operative time (min) | 61 | (50–90) | 62.5 | (52.3–84) | 61 | (46–90) | 0.917 |
| Resection weight (g) | 29 | (20–50) | 34 | (25.5–50.5) | 26 | (14.5–46) | 0.253 |
| Hemoglobin decrease (g/dL) | 1 | (0.4–1.65) | 1.15 | (0.5–1.8) | 1 | (0.3–1.4) | 0.186 |
| Indwelling catheter (days) | 5 | (4–6) | 5 | (4–6) | 5 | (4–6) | 0.574 |

IIEF-5: International Index of Erectile Function-5.

Data are shown as the median (IQR).

The follow-up data of this study are listed in Table 4. There was a significant improvement in urinary function in the comparison of preoperative, 3-, and 12-month follow-up data (median): Qmax (7.6, 12.9, 15.2 mL/s), PVR (50, 0, 0 mL), IPSS (20.5, 9, 6), and QoL (5, 2, 2). There were no significant differences in these parameters between groups 1 and 2 except for preoperative IPSS. PSA levels decreased from 5.8 (2.6–10.1) to 0.6 (0.4–1.3) ng/mL (89.2% decrease) at the 12-month follow-up. There was no significant change in the IIEF-5 scores in the comparison of preoperative, 3-, and 12-month follow-up data (median): (9, 7, 8). There was a slight but nonsignificant decrease in the IIEF-5 score in the comparison of preoperative, 3-, and 12-month follow-up in group 1 (13.5, 12.5, 13) and subgroup c (14, 13, 11). In contrast, there was a slight but significant increase in the IIEF-5 score in the comparison of preoperative and 12-month follow-up data in group 2 (5, 5, 6) and subgroup a (5, 5, 5). We also show in Fig 1 the proportion of patients who improved, remained unchanged, and worsened for each IIEF-5 score with reference to a previous study [20]. In group 1 and subgroup c, there were more patients with worsening IIEF-5 score than those with an improved score. Conversely, in group 2 and subgroup a, there were more patients with an improved IIEF-5 score than those with worsening score. We could not find a correlation between worsening of IIEF-5 scores (≥1 and ≥4) and urinary retention before TUEB, 5α reductase inhibitor therapy before TUEB, past history of hypertension, diabetes mellitus, cardiovascular disease, capsular perforation, operation time (≥61 or <61 minutes), and hemoglobin decrease (≥1.1 or <1.1 g/dL).

**Table 3. Complications (Clavien-Dindo classification).**

| Grade | Complication | Treatment | All men (n = 51) | | Patients with IIEF-5 ≥10 (n = 24) | | Patients with IIEF-5 <10 (n = 27) | |
|---|---|---|---|---|---|---|---|---|
| I | Capsular perforation | None | 3 | (5.9) | 2 | (8.3) | 1 | (3.7) |
| | Urinary retention | Recatheterization | 4 | (7.8) | 2 | (8.3) | 2 | (7.4) |
| | Hematuria | Prolonged bladder irrigation and hematoma evacuation | 3 | (5.9) | 1 | (4.2) | 2 | (7.4) |
| | Urinary incontinence | | | | | | | |
| | Stress | Oral administration | 1 | (2) | 1 | (4.2) | 0 | (0) |
| | Urge | Oral administration | 1 | (2) | 0 | (0) | 1 | (3.7) |
| II | Urinary tract infection | Antibiotics | 5 | (9.8) | 2 | (8.3) | 3 | (11.1) |
| IIIa | Urethral stricture | Dilation (bougie) | 3 | (5.9) | 0 | (0) | 3 | (11.1) |
| IIIb | Bladder neck sclerosis | Bladder neck incision | 1 | (2) | 0 | (0) | 1 | (3.7) |

IIEF-5: International Index of Erectile Function-5.

Data are shown as n (%).

**Table 4. Preoperative and follow-up data.**

| | Preoperative | | Follow-up (3 mo) | | Follow-up (12 mo) | | p Value | |
|---|---|---|---|---|---|---|---|---|
| | (n = 51) | | (n = 51) | | (n = 51) | | vs 3 mo** | vs 12 mo** |
| Qmax (mL/s)* | 7.6 | (5.8–10.9) | 12.9 | (9.1–17.6) | 15.2 | (10.5–19.9) | <0.001 | <0.001 |
| Group 1 | 6.9 | (5.4–10.6) | 12.5 | (10.1–21.35) | 16.4 | (12.7–22.4) | <0.001 | <0.001 |
| Group 2 | 8.4 | (6.5–11.2) | 13.3 | (8.6–16.9) | 14.8 | (10.2–18.2) | 0.0175 | <0.001 |
| p value (1 vs 2) | 0.174 | | 0.228 | | 0.549 | | | |
| PVR (mL)* | 50 | (3–107.8) | 0 | (0–10) | 0 | (0–7) | <0.001 | <0.001 |
| Group 1 | 53 | (22.5–94.8) | 0 | (0–5.5) | 0 | (0–1.5) | <0.001 | <0.001 |
| Group 2 | 42 | (3–107.8) | 0 | (0–16) | 4 | (0–9) | <0.001 | <0.001 |
| p value (1 vs 2) | 0.98 | | 0.409 | | 0.0967 | | | |
| IPSS | 20.5 | (16.8–28) | 9 | (6–13) | 6 | (4–11) | <0.001 | <0.001 |
| Group 1 | 18.5 | (14.3–22) | 8.5 | (4.5–11.5) | 6.5 | (2–10.5) | <0.001 | <0.001 |
| Group 2 | 24 | (18.3–30) | 9 | (7.5–15) | 6 | (5–11) | <0.001 | <0.001 |
| p value (1 vs 2) | 0.0266 | | 0.188 | | 0.421 | | | |
| QoL | 5 | (4–6) | 2 | (1–3) | 2 | (1–3) | <0.001 | <0.001 |
| Group 1 | 5 | (4–6) | 2 | (1–3.25) | 2 | (1–2) | <0.001 | <0.001 |
| Group 2 | 5 | (4–6) | 2 | (1–3) | 2 | (1–3) | <0.001 | <0.001 |
| p value (1 vs 2) | 0.991 | | 0.454 | | 0.121 | | | |
| PSA | 5.8 | (2.6–10.1) | | | 0.6 | (0.4–1.3) | | <0.001 |
| Group 1 | 7.4 | (3.9–10.6) | | | 0.6 | (0.4–1.2) | | <0.001 |
| Group 2 | 3.7 | (2–7.6) | | | 0.6 | (0.4–1.3) | | <0.001 |
| p value (1 vs 2) | 0.0713 | | | | 0.659 | | | |
| IIEF-5 | 9 | (5–12.5) | 7 | (5–14) | 8 | (5–14) | 0.647 | 0.956 |
| Group 1 (n = 24) | 13.5 | (10.8–15.5) | 12.5 | (7–15.3) | 13 | (8.8–16) | 0.0784 | 0.0851 |
| Group 2 (n = 27) | 5 | (5–6) | 5 | (5–6.5) | 6 | (5–7.5) | 0.105 | 0.0306 |
| p value (1 vs 2) | <0.001 | | <0.001 | | <0.001 | | | |
| IIEF-5 | 9 | (5–12.5) | 7 | (5–14) | 8 | (5–14) | 0.647 | 0.956 |
| Subgroup a (n = 23) | 5 | (5–5) | 5 | (5–6) | 5 | (5–7) | 0.0534 | 0.0122 |
| Subgroup b (n = 13) | 10 | (9–10) | 9 | (7–15) | 10 | (6–14) | 0.723 | 1 |
| Subgroup c (n = 9) | 14 | (13–15) | 13 | (10–14) | 11 | (8–14) | 0.172 | 0.159 |
| Subgroup d (n = 5) | 18 | (17–19) | 16 | (12–18) | 18 | (16–18) | 0.104 | 0.134 |
| Subgroup e (n = 1) | 23 | (23–23) | 23 | (23–23) | 23 | (23–23) | NaN | NaN |
| p value | <0.001 | | <0.001 | | <0.001 | | | |

Group 1: Patients with IIEF-5 ≥10 (n = 24), Group 2: Patients with IIEF-5 <10 (n = 27), Subgroup a: Patients with IIEF-5 Severe (5–7) (n = 23), Subgroup b: Patients with IIEF-5 Moderate (8–11) (n = 13), Subgroup c: Patients with IIEF-5 Mild to moderate (12–16) (n = 9), Subgroup d: Patients with IIEF-5 Mild (17–21) (n = 5), Subgroup e: Patients with IIEF-5 No erectile dysfunction (22–25) (n = 1), NaN: not a number. Other abbreviations as in Table 1.

Data are shown as median (interquartile range).

* Expect patients with urinary retention before operation.

** Compared with preoperative data.

## Discussion

HoLEP and thulium laser transurethral enucleation of the prostate (ThuLEP) are well-established enucleation procedures for patients with BPO [21]. However, in our experience, TUEB is relatively easy to learn and has almost equivalent safety and efficacy compared with laser enucleation surgeries such as HoLEP and ThuLEP [7, 21]. Because there are few data on how TUEB influences EF, unlike data on major transurethral surgery for BPO, we performed this prospective study to evaluate the influence of TUEB on EF.

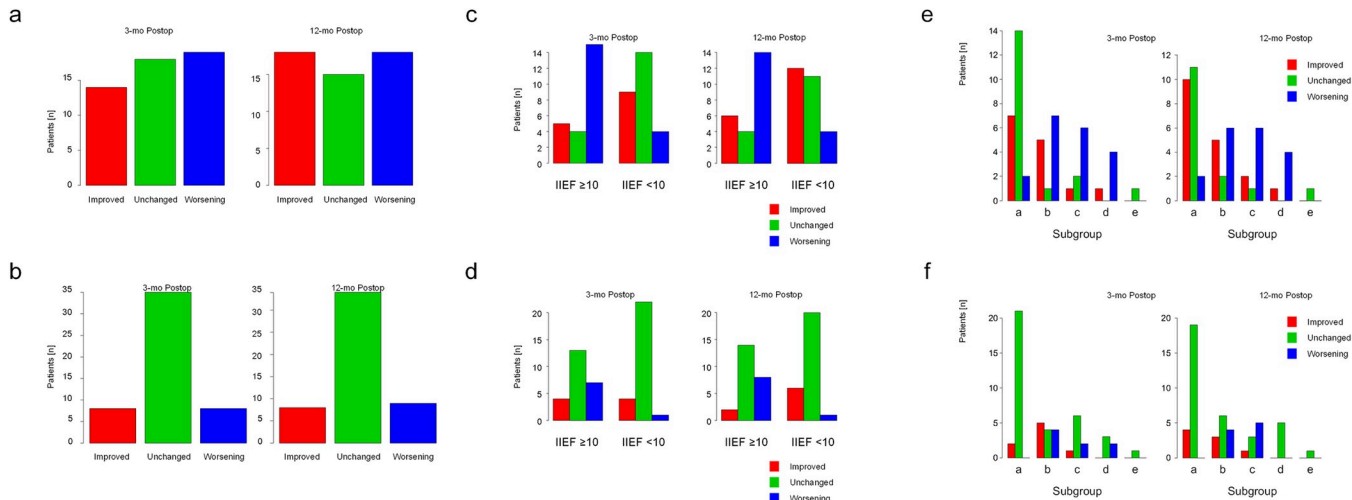

**Fig 1. Outcomes according to IIEF-5 score at 3- and 12-month follow-up after TUEB in patients with BPO.** (a) Outcomes based on changes in the IIEF-5 score ≥1. (b) Outcomes based on changes in the IIEF-5 score ≥4. (c) Outcomes based on changes in the IIEF-5 score ≥1 in patients with an IIEF-5 score ≥10 (group 1, left side of chart) and IIEF-5 score <10 (group 2, right side of chart). (d) Outcomes based on changes in the IIEF-5 score ≥4 in patients with an IIEF-5 score ≥10 (group 1, left side of chart) and an IIEF-5 score <10 (group 2, right side of chart). (e) Outcomes based on changes in the IIEF-5 score ≥1 in patients with an IIEF-5 score 5–7 (subgroup a), IIEF-5 score 8–11 (subgroup b), IIEF-5 score 12–16 (subgroup c), IIEF-5 score 17–21 (subgroup d), and IIEF-5 score 22–25 (subgroup e). (f) Outcomes based on changes in the IIEF-5 score ≥4 in patients with IIEF-5 scores as defined in panel (e). BPO: benign prostatic obstruction, IIEF: International Index of Erectile Function, TUEB: transurethral enucleation with bipolar.

In the present study, we found significant improvements of Qmax, PVR, IPSS, and QoL, and a decrease of PSA. Although Qmax at 12-month follow-up was slightly low, these results were almost comparable with previously reported results of transurethral enucleation surgeries such as HoLEP and ThuLEP (S1 Table) [22–28] and also almost equivalent with those of a prospective randomized trial between TUEB and TURP (S1 Table) [11]. One of the reasons for this slightly low Qmax might be that we did not exclude patients with neurogenic bladder, but we were unable to determine an exact reason because we did not perform pressure flow studies. Total rates of complications according to the Clavien-Dindo classification were 41.1% (23.5% [grade I], 9.8% [grade II], 5.9% [grade IIIa], and 2% [grade IIIb]) (Table 3). According to a recent review, complications in the transurethral enucleation surgeries showed the respective rates of stress and urge incontinence to be 0.7–28.6% and 0.6–48.1% and those of urethral stricture and bladder neck sclerosis to be 0.6–8.7% and 0.5–3.6% [29–31]. Although the rates of urethral stricture and bladder neck sclerosis might be relatively high (5.9% and 2%), we thought that the incidences of these complications were comparable with these previous studies [29–31].

We decided to assess EF with the IIEF-5 score because it is easy to use and understand compared with the original IIEF [19]. In addition, the IIEF-5 questionnaire has the advantage of being previously validated in Japanese, the language used by this study population. This study provided a prospective analysis of 51 patients who received TUEB and showed no significant influence on the IIEF-5 score in comparison with preoperative versus 3- and 12-month follow-up considering the entire population. In the groups, there was a slight but nonsignificant decrease in the IIEF-5 score in the comparison of preoperative, 3-, and 12-month follow-up data in group 1 and subgroup c. In contrast, there was a slight but significant increase in the IIEF-5 score in the comparison of preoperative and 12-month follow-up data in group 2 and subgroup a (Table 4).

Bruyere et al. reported that capsular perforation and a past history of cardiovascular disease were significant risk factors associated with EF after TURP [32]. In the present study, there

were 3 (5.9%) capsular perforations, which was higher than that of the previous study [7]. Contrary to the previous study, the IIEF-5 scores in all of the present patients did not worsen in the comparison of preoperative, 3-, and 12-month follow-up data: (5, 6, 6), (11, 7, 14), and (10, 15, 16), respectively. Because the number of cases in the present study was small, it is difficult to evaluate the effect of capsular perforation on EF. Capsular perforation occurred in the first 30 cases, which likely reflected the learning curve, and it will decrease with experience. There was a significant difference in the past history of cardiovascular disease between group 1 (IIEF-5 ≥10; n = 0) and group 2 (IIEF-5<10; n = 10). However compared to a previous study, the results showed a slight but nonsignificant decrease in the IIEF-5 score in group 1 and a slight but significant increase in the IIEF-5 score in group 2 [32]. We assessed whether cardiovascular disease influenced the IIEF-5 score, but it was not a significant factor worsening the IIEF-5 score in the present study. Unfortunately, we did not evaluate the severity of cardiovascular disease before TUEB in the present study.

Although the effect of the mechanism of transurethral surgery on EF is controversial, Akman et al. proposed that direct thermal injury to the erectile nerves leads to the worsening of EF [33]. Previous studies reported that the depth of coagulation was 0.14 mm with a wire loop for TURP, and the depths of penetration (i.e., the coagulation zone) were 0.4 mm with the holmium YAG laser used for HoLEP and 0.8 mm with the 532-nm (i.e., greenlight) laser used for PVP [20, 34, 35]. Considering the depth of coagulation, the coagulation effect of TUEB on EF might be similar to that of HoLEP. In recent years, several reports have been published regarding transurethral bipolar enucleation with a button electrode using it to coagulate bleeding vessels during the enucleation of prostate lobes [34, 36, 37]. However, the depth of coagulation was 2.4 mm with the button electrode for this surgery [20]. Although these reports mentioned that this surgery had no significant effect on EF, the coagulation effect of TUEB on EF might be smaller compared with this surgery because the depth of coagulation was considered to be shallow.

Akman et al. also proposed that thermal injury might have more impact at the apex than at the base [33]. They mentioned that the depth of erectile nerves was 1.5 mm at the apex and 3 mm at the base [33]. We could not evaluate the thermal effect on the procedure at the apex because we did not record the energy used such as that of laser energy. However, the basic procedure for HoLEP and TUEB is enucleation, which mechanically removes prostate adenoma between the adenoma and the capsule [7, 38]. These processes might result in a lower effect of thermal damage on EF compared with resection (TURP) or vaporization (PVP) [20].

The present study showed a slight but significant increase in the IIEF-5 score between preoperative and 12-month follow-up values in group 2 and subgroup a. A randomized controlled study comparing the influences of sexual function after HoLEP and TURP reported nonsignificant improvement in EF [38]. Several studies in which preoperative oral therapy including alpha-blocker or 5 alpha-reductase inhibitor was discontinued showed a slight contribution to the improvement of EF [11, 39]. However, we found no significant relationship between the change in IIEF-5 score and discontinuance of 5 alpha-reductase inhibitor following TUEB in the present study. Li et al. noted that the benefit of relief from BPO could counteract the negative effect of surgery on EF, leading to unchanged EF postoperatively [14]. Further studies are needed to investigate the effects of the discontinuance of preoperative oral therapy and surgery on EF.

Soans et al. found that patients with good preoperative EF may have worsened EF, and patients with severe preoperative EF may have improved EF [12]. The present study showed no significant influences on EF considering the overall population. In the groups, however, there was a slight but nonsignificant worsening of EF in group 1 and subgroup c and a slight but significant improvement in EF in group 2 and subgroup a. The nonsignificant worsening

of EF in subgroup c might be derived from not only the surgery itself but also aging or comorbidities, but we could not speculate as to the exact reason because the number or patients was small. We infer that the influence of TUEB on EF is minor in subgroups d and e, but again, the number of patients was too small. When patients with BPO are counseled prior to surgery, these findings might be an important factor.

This study has some limitations. Our sample size was rather small, and we could not find significant factors for a worsening IIEF-5 after TUEB. Second, an indwelling catheter was used for a relatively long time. We kept the urethral catheter in place for a longer period than usual to rest the bladder after TUEB to avoid temporary urinary retention or hemorrhage [7]. As well, a previous study reported that in a case series, TUEB required a shorter catheterization time (44.9 hours) [10]. Third, we could not compare this study cohort with randomized studies of other surgery such as monopolar or bipolar TURP. Fourth, most of the patients in this study had severe to moderate erectile dysfunction, and it was difficult for us to assess the influence on patients with mild to no erectile dysfunction. Indeed, because we have to consider the possibility of worsening of EF in subgroup c, we infer that the influence of TUEB on EF is minor in subgroups d and e. Fifth, the follow-up period of this study was only 12 months after TUEB. However, a previous study mentioned that changes in EF more than one year after surgery were derived from aging and comorbidities rather than the surgery itself [40]. Sixth, this study was a single-center non-randomized study. However, we think TUEB is an almost equally effective and safe surgery compared to transurethral laser enucleation surgeries such as HoLEP and ThuLEP [21]. Finally, this study includes relatively smaller-sized prostates, even though the great advantage of enucleation surgery is achieved in large prostates of over 100 mL [5]. A prospective multicenter study will be appropriate to assess the influence of other forms of transurethral surgery and TUEB on EF in patients with BPO.

## Conclusion

TUEB was effective and safe surgery for patients with BPO. The present study showed no significant influence of TUEB on EF in patients with BPO.

## Supporting information

**S1 Table. Prospective randomized trials of enucleation surgery of the prostate.**
(XLSX)

**S2 Table. Supporting information data set.**
(XLSX)

## Author Contributions

**Conceptualization:** Yasuyuki Kobayashi, Hiroki Arai.

**Data curation:** Yasuyuki Kobayashi, Hiroki Arai.

**Formal analysis:** Yasuyuki Kobayashi.

**Investigation:** Hiroki Arai.

**Project administration:** Yasuyuki Kobayashi.

**Supervision:** Masahito Honda.

**Writing – original draft:** Yasuyuki Kobayashi.

**Writing – review & editing:** Yasuyuki Kobayashi.

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
