## [Decision Letter · Decision Letter 0]

22 Mar 2022

PONE-D-21-39626Influence of transurethral enucleation with bipolar of the prostate on erectile function: Prospective analysis of 51 patients at 12-month follow-upPLOS ONE

Dear Dr. Kobayashi,

Thank you for submitting your manuscript to PLOS ONE. After careful consideration, we feel that it has merit but does not fully meet PLOS ONE’s publication criteria as it currently stands. Therefore, we invite you to submit a revised version of the manuscript that addresses the points raised during the review process.

ACADEMIC EDITOR:I hope you can adapt your manuscript as the reviewers requested. My advice is also to write your conclusin in a more modest way. ==============================

We look forward to receiving your revised manuscript.

Kind regards,

Peter F.W.M. Rosier, M.D. PhD

Academic Editor

PLOS ONE

Journal Requirements:

Reviewers' comments:

Reviewer's Responses to Questions

**Comments to the Author**

1. Is the manuscript technically sound, and do the data support the conclusions?

Reviewer #1: Yes

Reviewer #2: Partly

Reviewer #3: Partly

2. Has the statistical analysis been performed appropriately and rigorously? 

Reviewer #1: Yes

Reviewer #2: Yes

Reviewer #3: Yes

3. Have the authors made all data underlying the findings in their manuscript fully available?

Reviewer #1: Yes

Reviewer #2: Yes

Reviewer #3: Yes

4. Is the manuscript presented in an intelligible fashion and written in standard English?

Reviewer #1: Yes

Reviewer #2: Yes

Reviewer #3: Yes

5. Review Comments to the Author

Reviewer #1: I read the manuscript entitled:" Influence of transurethral enucleation with bipolar of the prostate on erectile function: Prospective analysis of 51 patients at 12-month follow-up"

All parts were written well but there is a concern about selection bias. All patients had erectile dysfunction before surgery (moderate as a whole or mild to moderate in group a and even severe in group b).

It is probable if the authors select patients with good erectile function maybe they could find any effect on erectile function.

Reviewer #2: The authors must be congratulated for their intent to evaluate the influence of TUEB on EF in patients with BPO.

However, the study shows all the results of a prospective database. Therefore, either the objective must be changed, or the results have to be adjusted, focusing in the proposed objective.

That being said, I would recommend to devide the patients according to the previously described ED classification based on the IIEF-5 score: severe (5-7), moderate (8-11), mild to moderate (12-16), mild (17-21), and no ED (22-25).

Reviewer #3: 1. Are you comparing your data to HoLEP and enucleation techniques or all prostate reducing procedures?

2. Advantages of using Bipolar enucleation versus well established enucleation techniques such as HoLEP and ThuLEP?

3. The data needs to be evaluated against TURP and thus a randomized prospective trial between the two would be necessary since the size of the gland appears to be in TURP category.

4. Many limitations to your study including one institution, size limitation, no comparison to other modalities, and too many patients with ED before the surgery.

5. Bipolar enucleation has not been established against TURP or other enucleation procedure.

6. Long catheterizations post op is concerning. By the way, in the discussion, ureteral catheterization should be changed to urethral catheterization.

7. The big advantage of enucleation techniques are for large glands usually over 100 grams, all the prostates in this study were less. What was the impetus for bipolar enucleation in this study?

8 I would suggest a re-write and compare your data only to enucleation procedures from a previously published randomized prospective trial. The flow the paper should be better and more concise. Your goal appears to suggest Bipolar enucleation can be performed with minimal effects on ED and with good outlet outcomes.

6. PLOS authors have the option to publish the peer review history of their article (what does this mean?). If published, this will include your full peer review and any attached files.

Reviewer #1: **Yes: **Farzad Allameh

Reviewer #2: No

Reviewer #3: No

---

## [Author Response · Author response to Decision Letter 0]

5 May 2022

Replies to the Editor and Reviewers

Dear Dr. Kobayashi,

Thank you for submitting your manuscript to PLOS ONE. After careful consideration, we feel that it has merit but does not fully meet PLOS ONE’s publication criteria as it currently stands. Therefore, we invite you to submit a revised version of the manuscript that addresses the points raised during the review process.

ACADEMIC EDITOR:

I hope you can adapt your manuscript as the reviewers requested. My advice is also to write your conclusin in a more modest way.

1→Thank you very much for your suggestion. I changed the sentence to make it easier to understand. We revised the text to reflect this change (Conclusion section, page 29, line 398).

2→Thank you very much for your advice.

I change the file name as follows:

Fig. 1→Fig1

We combined 6 figures into 1 figure in Fig. 2 and added panels e and f to panels a-d.

Although not pointed out, is it possible to make the following corrections?

3→We deleted the abbreviation “n.s.: not significant” in Table 2 and now show the actual p values.

Upon re-submitting your revised manuscript, please upload your study’s minimal underlying data set as either Supporting Information files

4→Thank you very much for your advice.

I upload our study’s minimal underlying data set as a Supporting Information file (file name: S2_Table).

Your ethics statement should only appear in the Methods section of your manuscript.

5→Thank you very much for your advice.

I deleted the “Ethics approval and consent to participate” section.

Reviewer #1: I read the manuscript entitled:" Influence of transurethral enucleation with bipolar of the prostate on erectile function: Prospective analysis of 51 patients at 12-month follow-up" All parts were written well but there is a concern about selection bias. All patients had erectile dysfunction before surgery (moderate as a whole or mild to moderate in group a and even severe in group b). It is probable if the authors select patients with good erectile function maybe they could find any effect on erectile function.

6→Thank you very much for your comments and suggestion. We classified the patients according to their preoperative IIEF-5 score into subgroup a: severe (IIEF-5 5-7, n=23), subgroup b: moderate (IIEF-5 8-11, n=13), subgroup c: mild to moderate (IIEF-5 12-16, n=9), subgroup d: mild (IIEF-5 17-21, n=5), and subgroup e: no ED (IIEF-5 22-25, n=1). There was a slight but nonsignificant decrease in the IIEF-5 score in the comparison of preoperative, 3-, and 12-month follow-up data in subgroup c (14, 13, 11) but no remarkable changes in subgroups d (18, 16, 18) and e (23, 23, 23). Although we cannot say for sure because of the small number of patients, we have to consider the possibility of worsening of EF in subgroup c but we infer that the influence of TUEB on EF is minor in subgroups d and e. We revised the text to reflect these results (Abstract section, page 2, line 31; Material and methods section, page 9, lines 146, 152; Results section, page 15, line 203/page 16, lines 209, 225; and Discussion section, page 23, line 292/page 27, lines 356, 358/page 28, line 376) and revised Fig 2 and Table 4.

Reviewer #2: The authors must be congratulated for their intent to evaluate the influence of TUEB on EF in patients with BPO. However, the study shows all the results of a prospective database. Therefore, either the objective must be changed, or the results have to be adjusted, focusing in the proposed objective. That being said, I would recommend to devide the patients according to the previously described ED classification based on the IIEF-5 score: severe (5-7), moderate (8-11), mild to moderate (12-16), mild (17-21), and no ED (22-25).

7→Thank you very much for your suggestion. We classified the patients according to your valuable advice as follows: subgroup a: severe (IIEF-5 5-7, n=23), subgroup b: moderate (IIEF-5 8-11, n=13), subgroup c: mild to moderate (IIEF-5 12-16, n=9), subgroup d: mild (IIEF-5 17-21, n=5), and subgroup e: no ED (IIEF-5 22-25, n=1). There was a slight but significant increase in the IIEF-5 score in the comparison of preoperative and 12-month follow-up data in subgroup a. There was a slight but nonsignificant decrease in the IIEF-5 score in the comparison of preoperative, 3-, and 12-month follow-up data in subgroup c (14, 13, 11) but no remarkable changes in subgroups d (18, 16, 18) and e (23, 23, 23). Although we cannot say for sure because of the small number of patients, we have to consider the possibility of worsening of EF in subgroup c but we infer that the influence of TUEB on EF is minor in subgroups d and e. We revised the text to reflect these results (Abstract section, page 2, line 31/page 3, line 45; Material and methods section, page 9, lines 146, 152; Results section, page 15, lines 203, 206/page 16, lines 209, 210; and Discussion section, page 23, lines 292, 294/page 26, line 341/page 27, lines 356, 357, 358/page 28, line 376) and revised Fig 2 and Table 4.

Reviewer #3: 1. Are you comparing your data to HoLEP and enucleation techniques or all prostate reducing procedures?

8→Thank you very much for your question. We compared our data with previously reported results of transurethral enucleation surgery such as HoLEP and ThuLEP. Although Qmax at 12-month follow-up was slightly low and the rate of urinary tract infection was relatively high, we think our data were almost comparable with those of previous studies. We revised the text to reflect these results (Discussion section, page 21, lines 252, 258/page 22, lines 264, 268, 269) and added Supplementary Table 1.

2. Advantages of using Bipolar enucleation versus well established enucleation techniques such as HoLEP and ThuLEP?

9→Thank you very much for your question. We think TUEB is relatively easy to learn, and we are able to control intraoperative bleeding with ease. We also think that we can precisely enucleate the adenoma from the capsule, which might reduce injury to the sphincter. We revised the text to reflect your suggestion (Discussion section, page 20, line 241).

3. The data needs to be evaluated against TURP and thus a randomized prospective trial between the two would be necessary since the size of the gland appears to be in TURP category.

10→Thank you very much for your suggestion. We compared our data with a prospective study between TUEB and TURP. Although Qmax at 12-month follow-up was slightly low, we think our data were almost comparable with those of this study. We revised the text to reflect this result (Discussion section, page 21, lines 255, 258) and added Supplementary Table 1.

4. Many limitations to your study including one institution, size limitation, no comparison to other modalities, and too many patients with ED before the surgery.

11→Thank you very much for your comment. We revised the limitations to reflect your suggestion (Discussion section, page 28, lines 376, 381, 384).

5. Bipolar enucleation has not been established against TURP or other enucleation procedure.

12→Thank you very much for your comment. We revised the text to reflect your suggestion (Introduction section, page 4, line 65).

6. Long catheterizations post op is concerning. 

13→Thank you very much for your comment. We think that we might be able to remove the catheter earlier. A previous study reported that TUEB required a shorter catheterization time. We revised the text to reflect your suggestion (Discussion section, page 27, line 368).

By the way, in the discussion, ureteral catheterization should be changed to urethral catheterization.

14→Thank you very much for pointing out our mistake. We corrected this error (Discussion section, page 28. line 370).

7. The big advantage of enucleation techniques are for large glands usually over 100 grams, all the prostates in this study were less. What was the impetus for bipolar enucleation in this study?

15→Thank you very much for your comment and question. This study included 10 patients with prostate size over 100 mL, and we were able to perform surgery without any problems using the same procedure. We revised the text to reflect your suggestion (Discussion section, page 28, line 384).

8 I would suggest a re-write and compare your data only to enucleation procedures from a previously published randomized prospective trial. The flow the paper should be better and more concise. Your goal appears to suggest Bipolar enucleation can be performed with minimal effects on ED and with good outlet outcomes.

16→Thank you very much for your comment. As mentioned, we compared our data with a randomized prospective trial of transurethral enucleation surgery. Although Qmax at 12-month follow-up was slightly low and the rate of urinary tract infection was relatively high, we think our data were almost comparable with these studies. We might be able to suggest that TUEB can be performed with minimal effects on ED and with good outlet outcomes. We revised the text to reflect this result (Discussion section, page 21, lines 252/258; page 22, lines 264/268, 269) and added Supplementary Table 1.

Word limit in the Abstract section

17→Dear editor,

I revised the Abstract section according to the comments of Reviewers #1 and #2. As a result, the word count now exceeds 300 words (304 words). I hope this will not be a problem.

While revising your submission, please upload your figure files to the Preflight Analysis and Conversion Engine (PACE) digital diagnostic tool

→Thank you very much for your advice. I checked the figure files with PACE.

---

## [Decision Letter · Decision Letter 1]

9 Jun 2022

PONE-D-21-39626R1Influence of transurethral enucleation with bipolar of the prostate on erectile function: Prospective analysis of 51 patients at 12-month follow-upPLOS ONE

Dear Dr. Kobayashi,

Thank you for submitting your manuscript to PLOS ONE. After careful consideration, we feel that it has merit but does not fully meet PLOS ONE’s publication criteria as it currently stands. Therefore, we invite you to submit a revised version of the manuscript that addresses the points raised during the review process.

ACADEMIC EDITOR: Can you revise again, with help of the reviewers comments?

We look forward to receiving your revised manuscript.

Kind regards,

Peter F.W.M. Rosier, M.D. PhD

Academic Editor

PLOS ONE

Journal Requirements:

Reviewers' comments:

Reviewer's Responses to Questions

**Comments to the Author**

1. If the authors have adequately addressed your comments raised in a previous round of review and you feel that this manuscript is now acceptable for publication, you may indicate that here to bypass the “Comments to the Author” section, enter your conflict of interest statement in the “Confidential to Editor” section, and submit your "Accept" recommendation.

Reviewer #2: (No Response)

Reviewer #3: All comments have been addressed

2. Is the manuscript technically sound, and do the data support the conclusions?

Reviewer #2: Yes

Reviewer #3: Partly

3. Has the statistical analysis been performed appropriately and rigorously? 

Reviewer #2: Yes

Reviewer #3: Yes

4. Have the authors made all data underlying the findings in their manuscript fully available?

Reviewer #2: (No Response)

Reviewer #3: Yes

5. Is the manuscript presented in an intelligible fashion and written in standard English?

Reviewer #2: Yes

Reviewer #3: Yes

6. Review Comments to the Author

Reviewer #2: Thanks for addressing all the raised questions.

Some minor issues:

Abstract:

Material and methods:

The groups identification is quite complex. There is initially a GROUP A and GROUP B. Then, there are subgroups a, b, c, d and e. This leads to misunderstandings and is hard to interpretate. Maybe you could use groups 1 and 2 and subgroups a to e.

I could not understand the following sentence: “Data are displayed as median or median (interquartile range) or (preoperative, at 3-month follow-up, at 12-month follow-up).”. In the abstract and in the method section.

Methods

Line 142: Please describe the groups in a new paragraph

Line 138: statistical analysis description should be the last paragraph of the methods section.

Results/ Objectives:

Once again, I believe the results and discussion deal with a lot of things that are not mentioned in the objective of the study!

The aim of the study was: “Therefore, we conducted a single-center, prospective study to evaluate the influence of TUEB on EF in 51 patients with BPO.”

The following topics have nothing to do with the aim of the study, unless you evaluate if theses complications could interfere with the post-operative erectile function.

Line 172: correlation between TZ volume and resection weight

Line 175: patients with transient urinary retention after catheter removal, urinary retention resolved spontaneously without oral administration of medication

Line: 177: patients with hematuria

Line 179: urethral structure and bladder neck sclerosis

Table 3 is enough to show the complications due to the procedure.

Figure 1 is not necessary.

Discussion:

Regarding complications, instead of citing 7 studies, I would cite

1) for incontinence (line 254):

- Castellani D, Rubilotta E, Fabiani A, Maggi M, Wroclawski M, Teoh JYC, Pirola GM, Gubbioti M, Pavia MP, Gomez Sancha F, Galosi AB, Gauhar V. Correlation between transurethral interventions and their influence on type and duration of postoperative urinary incontinence: results from a systematic review and meta-analysis of comparative studies. J Endourol. 2022 May 19. doi: 10.1089/end.2022.0222.

2) For urethral stricture (line 255):

- Pirola GM, Castellani D, Lim EJ, Wroclawski ML, Le Quy Nguyen D, Gubbiotti M, Rubilotta E, Chan VW, Corrales M, Rojo EG, Herrmann TRW, Teoh JY, Gauhar V. Urethral stricture following endoscopic prostate surgery: a systematic review and meta-analysis of prospective, randomized trials. World J Urol. 2022 Feb 13. doi: 10.1007/s00345-022-03946-z.

3) For bladder neck sclerosis (line 255):

- Castellani D, Wroclawski ML, Pirola GM, Gauhar V, Rubilotta E, Chan VW, Cheng BK, Gubbiotti M, Galosi AB, Herrmann TRW, Teoh JY. Bladder neck stenosis after transurethral prostate surgery: a systematic review and meta-analysis. World J Urol. 2021 Nov;39(11):4073-4083. doi: 10.1007/s00345-021-03718-1.

Lines 259 – 268 (TZ volume and MRI) does not correlate to the topic of the study. In my opinion, it should be deleted.

Line 365: Regarding the safety and efficacy of TUEB, I would cite the metanalysis that compared laser versus non-laser enucleation procedures and did not find any significant differences between the techniques (Wroclawski ML, Teles SB, Amaral BS, Kayano PP, Cha JD, Carneiro A, Alfer W Jr, Monteiro J Jr, Gil AO, Lemos GC. A systematic review and meta-analysis of the safety and efficacy of endoscopic enucleation and non-enucleation procedures for benign prostatic enlargement. World J Urol. 2020 Jul;38(7):1663-1684. doi: 10.1007/s00345-019-02968-4.)

Line 375: Enucleation is a procedure indicated for prostates of any size according to the AUA guidelines and for prostates over 30g according to the EAU guidelines. So, no need to justify your approach.

Reviewer #3: This is one section that should be revised.

"HoLEP is a well-established enucleation surgery in patients with BOO

233 [5]. However, HoLEP requires much experience and appropriate endoscopic

234 skills to avoid complications [21], whereas TUEB is relatively easy to learn

235 [7]. We can coagulate any bleeding immediately during adenoma enucleation

236 because the TUEB loop is equipped with a standard wire loop. In addition, we

237 can precisely enucleate the adenoma from the capsule with the TUEB loop,

238 which might reduce injury to the sphincter at the apex [11]."

This section should be re-written. This is truly an opinion from the surgeons doing bipolar enucleations. Learning curve has been defined with other procedures over longer time period at different institutions. Reproducibility is an important part of learning curve and multi-institutional trials need to be done to appropriately comment on this.

Phrasing should be not so presumptive. Accuracy with Holmium and Thulium of bleeders are also easily accomplished with experience. I realize your enthusiasm of this technique, however, you need to tone it down throughout the discussion and introduction. Maybe try the phrase, "In our experience,..." to phrase some the presumptive statements. In addition, please focus on erectile function as the title would suggest.

7. PLOS authors have the option to publish the peer review history of their article (what does this mean?). If published, this will include your full peer review and any attached files.

Reviewer #2: No

Reviewer #3: No

---

## [Author Response · Author response to Decision Letter 1]

26 Jun 2022

Replies to the Editor and Reviewers

Dear Dr. Kobayashi,

Thank you for submitting your manuscript to PLOS ONE. After careful consideration, we feel that it has merit but does not fully meet PLOS ONE’s publication criteria as it currently stands. Therefore, we invite you to submit a revised version of the manuscript that addresses the points raised during the review process.

ACADEMIC EDITOR:

Can you revise again, with help of the reviewers comments?

→We are grateful for this additional opportunity to revise our paper. We thank you and the reviewers for the time and effort spent providing fine feedback to strengthen our paper. We have revised and submitted our updated manuscript.

Journal Requirements:

→Thank you very much for your suggestion. We have checked all articles cited in the references, and none show any indication that they have been retracted.

Reviewers' comments:

Reviewer's Responses to Questions

Reviewer #2: Thanks for addressing all the raised questions.

Some minor issues:

Abstract:

Material and methods:

The groups identification is quite complex. There is initially a GROUP A and GROUP B. Then, there are subgroups a, b, c, d and e. This leads to misunderstandings and is hard to interpretate. Maybe you could use groups 1 and 2 and subgroups a to e.

1→Thank you very much for your valuable advice, which has made the results easier to understand. We revised the two main group names to Group 1 and Group 2 throughout the manuscript and in Tables 1 and 4.

I could not understand the following sentence: “Data are displayed as median or median (interquartile range) or (preoperative, at 3-month follow-up, at 12-month follow-up).”. In the abstract and in the method section.

2→Thank you very much for your advice. We deleted the confusing second half of the sentence to make it easier to understand. We revised the text to reflect this change in the Abstract, page 2, line 33, and the Material and Methods section, page 9, line 149.

Methods

Line 142: Please describe the groups in a new paragraph

3→Thank you very much for your suggestion. We revised the text to reflect your suggestion (Material and Methods section, page 8, line 143).

Line 138: statistical analysis description should be the last paragraph of the methods section.

4→Thank you very much for your suggestion. All text related to statistical analysis is included in the last paragraph of the Material and Methods section (page 9, lines 149-161).

Results/ Objectives:

Once again, I believe the results and discussion deal with a lot of things that are not mentioned in the objective of the study!

The aim of the study was: “Therefore, we conducted a single-center, prospective study to evaluate the influence of TUEB on EF in 51 patients with BPO.”

The following topics have nothing to do with the aim of the study, unless you evaluate if theses complications could interfere with the post-operative erectile function.

Line 172: correlation between TZ volume and resection weight

5→Thank you very much for your advice, which has improved the flow of this paper and made it more concise. We deleted the sentence you pointed out and another related sentence (Material and Methods section, page 9, line 152 and Results section, page 13, line 179).

Line 175: patients with transient urinary retention after catheter removal, urinary retention resolved spontaneously without oral administration of medication

6→We deleted the sentence you pointed out (Results section, page 13, line 181).

Line: 177: patients with hematuria

7→We deleted the sentence you pointed out (Results section, page 13, line 183).

Line 179: urethral structure and bladder neck sclerosis

8→We deleted the sentence you pointed out (Results section, page 13, line 185).

Table 3 is enough to show the complications due to the procedure.

Figure 1 is not necessary.

9→Thank you. We deleted Figure 1. Previous Figure 2 is now Figure 1.

Discussion:

Regarding complications, instead of citing 7 studies, I would cite

1) for incontinence (line 254):

- Castellani D, Rubilotta E, Fabiani A, Maggi M, Wroclawski M, Teoh JYC, Pirola GM, Gubbioti M, Pavia MP, Gomez Sancha F, Galosi AB, Gauhar V. Correlation between transurethral interventions and their influence on type and duration of postoperative urinary incontinence: results from a systematic review and meta-analysis of comparative studies. J Endourol. 2022 May 19. doi: 10.1089/end.2022.0222.

10→Thank you for alerting us to these three very useful papers. We revised the text to reflect your suggestions (Discussion section, page 22, line 279).

2) For urethral stricture (line 255):

- Pirola GM, Castellani D, Lim EJ, Wroclawski ML, Le Quy Nguyen D, Gubbiotti M, Rubilotta E, Chan VW, Corrales M, Rojo EG, Herrmann TRW, Teoh JY, Gauhar V. Urethral stricture following endoscopic prostate surgery: a systematic review and meta-analysis of prospective, randomized trials. World J Urol. 2022 Feb 13. doi: 10.1007/s00345-022-03946-z.

11→We revised the text to reflect your suggestion (Discussion section, page 22, line 281).

3) For bladder neck sclerosis (line 255):

- Castellani D, Wroclawski ML, Pirola GM, Gauhar V, Rubilotta E, Chan VW, Cheng BK, Gubbiotti M, Galosi AB, Herrmann TRW, Teoh JY. Bladder neck stenosis after transurethral prostate surgery: a systematic review and meta-analysis. World J Urol. 2021 Nov;39(11):4073-4083. doi: 10.1007/s00345-021-03718-1.

12→We revised the text to reflect your suggestion (Discussion section, page 23, line 282).

Lines 259 – 268 (TZ volume and MRI) does not correlate to the topic of the study. In my opinion, it should be deleted.

13→We deleted the related sentences you pointed out (Discussion section, page 23, lines 286-295).

Line 365: Regarding the safety and efficacy of TUEB, I would cite the metanalysis that compared laser versus non-laser enucleation procedures and did not find any significant differences between the techniques (Wroclawski ML, Teles SB, Amaral BS, Kayano PP, Cha JD, Carneiro A, Alfer W Jr, Monteiro J Jr, Gil AO, Lemos GC. A systematic review and meta-analysis of the safety and efficacy of endoscopic enucleation and non-enucleation procedures for benign prostatic enlargement. World J Urol. 2020 Jul;38(7):1663-1684. doi: 10.1007/s00345-019-02968-4.)

14→Thank you for telling us about this very useful paper. We revised the text to reflect your suggestion (Discussion section, page 29, lines 394-396).

Line 375: Enucleation is a procedure indicated for prostates of any size according to the AUA guidelines and for prostates over 30g according to the EAU guidelines. So, no need to justify your approach.

15→Thank you very much for your advice. We deleted the sentences to reflect your suggestion (Discussion section, page 29, lines 398-400).

Reviewer #3: This is one section that should be revised.

"HoLEP is a well-established enucleation surgery in patients with BOO

233 [5]. However, HoLEP requires much experience and appropriate endoscopic

234 skills to avoid complications [21], whereas TUEB is relatively easy to learn

235 [7]. We can coagulate any bleeding immediately during adenoma enucleation

236 because the TUEB loop is equipped with a standard wire loop. In addition, we

237 can precisely enucleate the adenoma from the capsule with the TUEB loop,

238 which might reduce injury to the sphincter at the apex [11]."

This section should be re-written. This is truly an opinion from the surgeons doing bipolar enucleations. Learning curve has been defined with other procedures over longer time period at different institutions. Reproducibility is an important part of learning curve and multi-institutional trials need to be done to appropriately comment on this.

Phrasing should be not so presumptive. Accuracy with Holmium and Thulium of bleeders are also easily accomplished with experience. I realize your enthusiasm of this technique, however, you need to tone it down throughout the discussion and introduction. Maybe try the phrase, "In our experience,..." to phrase some the presumptive statements. In addition, please focus on erectile function as the title would suggest.

16→Thank you very much for your advice. We used the phrase "In our experience" to focus on the ease of learning TUEB and reduced any presumptiveness on our part. We also focused on erectile function in this paragraph (Discussion section, page 21, lines 251-259).

---

## [Decision Letter · Decision Letter 2]

25 Jul 2022

Influence of transurethral enucleation with bipolar of the prostate on erectile function: Prospective analysis of 51 patients at 12-month follow-up

PONE-D-21-39626R2

Dear Dr. Kobayashi,

We’re pleased to inform you that your manuscript has been judged scientifically suitable for publication and will be formally accepted for publication once it meets all outstanding technical requirements.

Kind regards,

Peter F.W.M. Rosier, M.D. PhD

Academic Editor

PLOS ONE

Additional Editor Comments (optional):

Reviewers' comments:

Reviewer's Responses to Questions

**Comments to the Author**

1. If the authors have adequately addressed your comments raised in a previous round of review and you feel that this manuscript is now acceptable for publication, you may indicate that here to bypass the “Comments to the Author” section, enter your conflict of interest statement in the “Confidential to Editor” section, and submit your "Accept" recommendation.

Reviewer #2: All comments have been addressed

Reviewer #3: All comments have been addressed

2. Is the manuscript technically sound, and do the data support the conclusions?

Reviewer #2: Yes

Reviewer #3: Yes

3. Has the statistical analysis been performed appropriately and rigorously? 

Reviewer #2: Yes

Reviewer #3: Yes

4. Have the authors made all data underlying the findings in their manuscript fully available?

Reviewer #2: Yes

Reviewer #3: Yes

5. Is the manuscript presented in an intelligible fashion and written in standard English?

Reviewer #2: Yes

Reviewer #3: Yes

6. Review Comments to the Author

Reviewer #2: (No Response)

Reviewer #3: (No Response)

7. PLOS authors have the option to publish the peer review history of their article (what does this mean?). If published, this will include your full peer review and any attached files.

Reviewer #2: No

Reviewer #3: No

---

## [Editor Report · Acceptance letter]

2 Aug 2022

PONE-D-21-39626R2 

Influence of transurethral enucleation with bipolar of the prostate on erectile function: Prospective analysis of 51 patients at 12-month follow-up 

Dear Dr. Kobayashi:

I'm pleased to inform you that your manuscript has been deemed suitable for publication in PLOS ONE. Congratulations! Your manuscript is now with our production department. 

Kind regards, 

on behalf of

Dr. Peter F.W.M. Rosier 

Academic Editor

PLOS ONE